# Structure of the ceramide-bound SPOTS complex

Jan-Hannes Schäfer [1,4], Carolin Körner[2,4], Bianca M. Esch[2], Sergej Limar[2], Kristian Parey [1,3], Stefan Walter [3], Dovile Januliene [1,3] ✉, Arne Moeller [1,3] ✉ & Florian Fröhlich [2,3] ✉

Sphingolipids are structural membrane components that also function in cellular stress responses. The serine palmitoyltransferase (SPT) catalyzes the rate-limiting step in sphingolipid biogenesis. Its activity is tightly regulated through multiple binding partners, including Tsc3, Orm proteins, ceramides, and the phosphatidylinositol-4-phosphate (PI4P) phosphatase Sac1. The structural organization and regulatory mechanisms of this complex are not yet understood. Here, we report the high-resolution cryo-EM structures of the yeast SPT in complex with Tsc3 and Orm1 (SPOT) as dimers and monomers and a monomeric complex further carrying Sac1 (SPOTS). In all complexes, the tight interaction of the downstream metabolite ceramide and Orm1 reveals the ceramide-dependent inhibition. Additionally, observation of ceramide and ergosterol binding suggests a co-regulation of sphingolipid biogenesis and sterol metabolism within the SPOTS complex.

Sphingolipids are essential membrane components in eukaryotes, composed of a sphingosine backbone with a fatty acid attached, and are further modified through the addition of various polar head groups. They are particularly abundant in the plasma membrane, contributing to its structural integrity and function[1]. In addition, sphingolipids act as signaling molecules, for example, in apoptosis and the immune response[2–4]. Sphingolipid metabolism is tightly regulated via multiple signals and additionally linked to sterol levels[5–7]. Imbalances in the levels of sphingolipids and sterols are implicated in a variety of human pathologies, including neurodegenerative diseases such as Niemann-Pick type C and childhood amyotrophic lateral sclerosis (ALS)[8,9].

Serine palmitoyltransferase (SPT) is the rate-limiting enzyme in the synthesis of sphingolipids. It catalyzes the transfer of a palmitoyl group to L-serine, yielding 3-ketosphinganine (3-KS)[10]. 3-KS is reduced to long-chain bases, which are further processed into ceramides and complex sphingolipids[11]. SPT is highly conserved across species and consists of two large catalytic subunits (in yeast: Lcb1 and Lcb2) and interacts with a small regulatory subunit (in yeast: Tsc3)[12–17]. Tsc3

modulates enzyme activity through various mechanisms, including allosteric regulation and protein-protein interactions[18].

Two recent structural studies of the human SPT showed that the enzyme acts as a homodimer with the two transmembrane helices of the human Lcb1 (SPTLC1) subunits swapped between dimers[19,20]. The small regulatory subunit ssSPTa provides an additional transmembrane helix, and the ORMDL3 protein is located in between the transmembrane helices of SPTLC1 and ssSPTa. The SPT structures revealed the mechanism of substrate recognition and fatty acid selectivity. Regulation of the SPT complex occurs through multiple input signals, including Orm proteins, which are co-purified with the SPT[21].

Orm proteins (ORMDL1/2/3 in mammalian cells, Orm1/2 in yeast cells) act as negative regulators of the SPT[21,22]. Mammalian ORMDL3 extends its N-terminus into the active site of SPT, thus inhibiting enzyme activity[19,20]. Yeast Orm proteins have extended N-termini that are not evolutionarily conserved, suggesting a different mechanism of regulation. In line, yeast Orm proteins are phosphorylated at the extended N-terminus by the Ypk kinases, leading to increased SPT

[1]Osnabrück University Department of Biology/Chemistry Structural Biology section, 49076 Osnabrück, Germany. [2]Osnabrück University Department of Biology/Chemistry Bioanalytical Chemistry section, 49076 Osnabrück, Germany. [3]Osnabrück University Center of Cellular Nanoanalytic Osnabrück (Cell-NanOs), 49076 Osnabrück, Germany. [4]These authors contributed equally: Jan-Hannes Schäfer, Carolin Körner. ✉e-mail: dovile.januliene@uos.de; arne.moeller@uos.de; florian.froehlich@uos.de

activity[23–25]. In addition, SPT activity is reduced in the presence of ceramides, the downstream metabolites of long-chain base/sphingosine synthesis. This regulation is proposed to depend on the presence of Orm proteins, but the molecular mechanism remains elusive[21,22,26].

In yeast, the SPT-Orm-Tsc3 complex (SPOT) harbors an additional partner, the PI4P phosphatase Sac1 (SPOTS complex)[27–29]. Sac1 has been proposed to modulate sphingolipid metabolism through its interaction with the SPOT complex. However, neither its binding mode nor its specific function within the SPOTS complex are known, but its deletion affects sphingolipid levels[30].

Here, we solved cryo-EM structures of the yeast SPOT complex in both monomeric and dimeric states and the monomeric SPOTS complex. The overall architecture of the individual subunits is almost indistinguishable from yeast to humans. A marked difference is the absence of the previously reported crossover helices at the protomer interface in our dimeric structure, which could explain why we were able to obtain monomeric SPOT complexes. Notably, the PI4P phosphatase Sac1 binds exclusively to the monomeric complex. Our data show that in yeast, Orm1 does not regulate SPT via insertion of its N-terminus into the active site but rather in conjunction with ceramide. We identified ceramide in all complexes, coordinated between Orm1 and Lcb2, blocking the SPT substrate channel. Furthermore, we revealed the presence of several ergosterol molecules in the monomeric complexes, suggesting that the SPOTS complex is a regulatory junction to coordinate sphingolipid and sterol levels. Together, we provide a structural basis for SPT regulation via multiple signals.

## Results

To unravel the architecture of the yeast SPOTS complex, we generated a *S. cerevisiae* strain co-expressing Lcb1, Lcb2, Tsc3, Orm1, and Sac1 (Sup. Fig. 1a) under the control of the inducible *GAL1* promoter. Lcb1 was internally FLAG-tagged after P9 to enable affinity purification while maintaining the functionality of Lcb1[14]. In addition, the three known phosphorylation sites S51, S52, and S53 of Orm1 were mutated to alanine to yield a non-phosphorylatable version (ORM1$^{AAA}$). We reasoned that this would stabilize the entire complex. These three serine residues are target sites for the regulatory yeast Ypk kinase, which upon phosphorylation, increases SPT activity[21,23]. We anticipated that the ORM1$^{AAA}$ mutant would render the SPOTS complex inactive; however, it still showed enzymatic activity of 45 nmol mg$^{-1}$ min$^{-1}$ and was sensitive to myriocin (Sup. Fig. 1f).

For cryo-EM studies, the *S. cerevisiae* SPOTS complex was solubilized in glyco-diosgenin (GDN) and purified by FLAG-based affinity chromatography (Sup. Fig. 1b). The purified complex was also subjected to mass spectrometric analysis, confirming the presence of all subunits with high sequence coverage (Sup. Fig. 1d). Multi-model single particle cryo-EM revealed three different compositions of the complex within one dataset, including a C2 symmetric SPOT dimer (Fig. 1a) and two SPOT monomers, among which one additionally contains the regulatory subunit Sac1 (SPOTS complex) (Fig. 1b, c).

### The architecture of the SPOT dimer

The SPOT dimer was refined to 3.4 Å resolution, with C2-symmetry imposed. Symmetry expansion of one protomer improved the resolution further to 3.0 Å (Sup. Tab.1 and Sup. Figs. 3, 4). As previously reported, Lcb1 and Lcb2 build the enzymatic core of the complex. In contrast to the previously suggested architecture of yeast Lcb1[13], only a single transmembrane helix (TM1) located at the N-terminal part of the protein is visible and anchors Lcb1 to the membrane, while Lcb2 is embedded in the membrane via an amphipathic helix (Fig. 2a, Sup. Fig. 6, Sup. Fig. 8). The regulatory subunit Tsc3 provides an additional membrane anchor through its single transmembrane helix and an amphipathic helix (Fig. 2a, Sup. Fig. 6, Sup. Fig. 8). Tsc3 does not interact with Lcb1, but it binds tightly to Lcb2 via an elongated N-terminal region that is not conserved in the human homolog (Fig. 2d,

Sup. Fig. 9e). Orm1 is positioned between the amphipathic helix of Lcb2 (Fig. 2g) and the TM1 helix of Lcb1 (Fig. 2i) but does not interact with Tsc3.

The general architecture of the SPOT complex is remarkably conserved from yeast to human (Sup. Fig. 9). However, our structure shows a different arrangement of the two Lcb1 transmembrane helices (TM1), which were previously reported to establish a crossover between the two protomers[19,20], leading to an extensive interface within the membrane (Sup. Fig. 7a). We do not observe such helix crossover for the yeast SPOT dimer. Nevertheless, the relative position of the Lcb1 transmembrane helix of one protomer superimposes with the corresponding crossover helix of the adjacent protomer in the human dimeric structure (Sup. Fig. 7b, c). In yeast SPOT, the protomer binding interface is established through interactions between Lcb1$^{K85}$ - Lcb2$^{N291}$ and Lcb1$^{N87}$ -

Lcb2$^{K289}$ and between the adjacent Lcb2$^{V284}$ - Lcb2$^{*R292}$ (Fig. 2c) in the cytosol-facing portion of the protein. The distance between the Lcb1 transmembrane helices of our dimer structure is increased from 13 to 28 Å (Sup. Fig. 7a), reminiscent of the previously reported ORMDL3-free SPT complex (PDB: 7K0I, 2.8 Å global RMSD).

The small human ssSPTa subunit has been shown to regulate fatty acid selectivity via the insertion of a methionine side chain in the substrate binding tunnel. The corresponding Tsc3 subunit in yeast harbors a methionine (M51) at a similar position between its transmembrane and amphipathic helix, but its side chain does not extend into the substrate access channel (Fig. 2g). Interestingly, Tsc3 has been reported to regulate the amino acid choice of the SPT rather than controlling fatty acid selectivity as shown for ssSPTa[18].

Superposition of ORMDL3 and Orm1 reveals very high structural conservation; only the regulatory N-terminal parts exhibit marked differences (Fig. 3c, Sup. Fig. 9c). In the human SPT complex, the N-terminal methionine of ORMDL3 reaches into the substrate binding tunnel in SPTLC2, resulting in negative modulation of SPT activity. This requires a sharp kink from residue V10 to N11. The corresponding asparagine N74 in yeast is preceded by P73, which introduces a sharp kink in the opposite direction and folds into a structured helix, docking tightly into a socket composed of Lcb2 (Figs. 2h, 3c). Of note, the yeast Orm proteins harbor additional 63 N-terminal amino acids that are not present in the human proteins (Sup. Fig. 9b, c). In our structure, the N-terminus of Orm1 latches onto the surface of Lcb1 before it interacts with its own C-terminus in an antiparallel beta-sheet (Fig. 3c). Consequently, the N-terminus cannot fulfill the same function that has been reported for the human SPT-ORMDL3, explaining the need for different modes of regulation that have been observed.

As shown for the SPT-ORMDL3 structures, also in our structure, the active site between Lcb1 and Lcb2 is populated by the cofactor pyridoxal 5′-phosphate (PLP) (Fig. 1a, Sup. Fig. 5b), which forms an internal aldimine with Lcb2-K366, essential for the catalysis of serine and acyl-CoA condensation reaction[31]. Likewise, the putative substrate access tunnel is gated by the conserved PATP loop of Lcb2/SPTLC2 (in yeast, amino acids 486–489, Fig. 3b). Below the PATP loop, at the interface between the amphipathic helix of Lcb2 (amino acids 58-85) and Orm1, we identified an elongated density (Figs. 1b, c, 2e and 3a, b, Sup. Fig. 4e, Sup. Fig. 5a), that was not detected in the human structures. Two long acyl chains and the lack of a prominent head group indicate the presence of a very long-chain fatty acid containing ceramide. In addition, we see a clear density for a GDN molecule, interacting with the acyl chains of the ceramide (Sup. Fig. 5d, Sup. Fig. 4e).

Since the elongated ceramide density exhibits signs of signal attenuation, suggesting mixed occupancy or flexibility, we employed the automated estimation of compositional heterogeneity using "OccuPy"[32], which supports our interpretation (Sup. Fig. 11). To further corroborate the presence of ceramide in our structure, the sample used for cryo-EM was subjected to lipid extraction

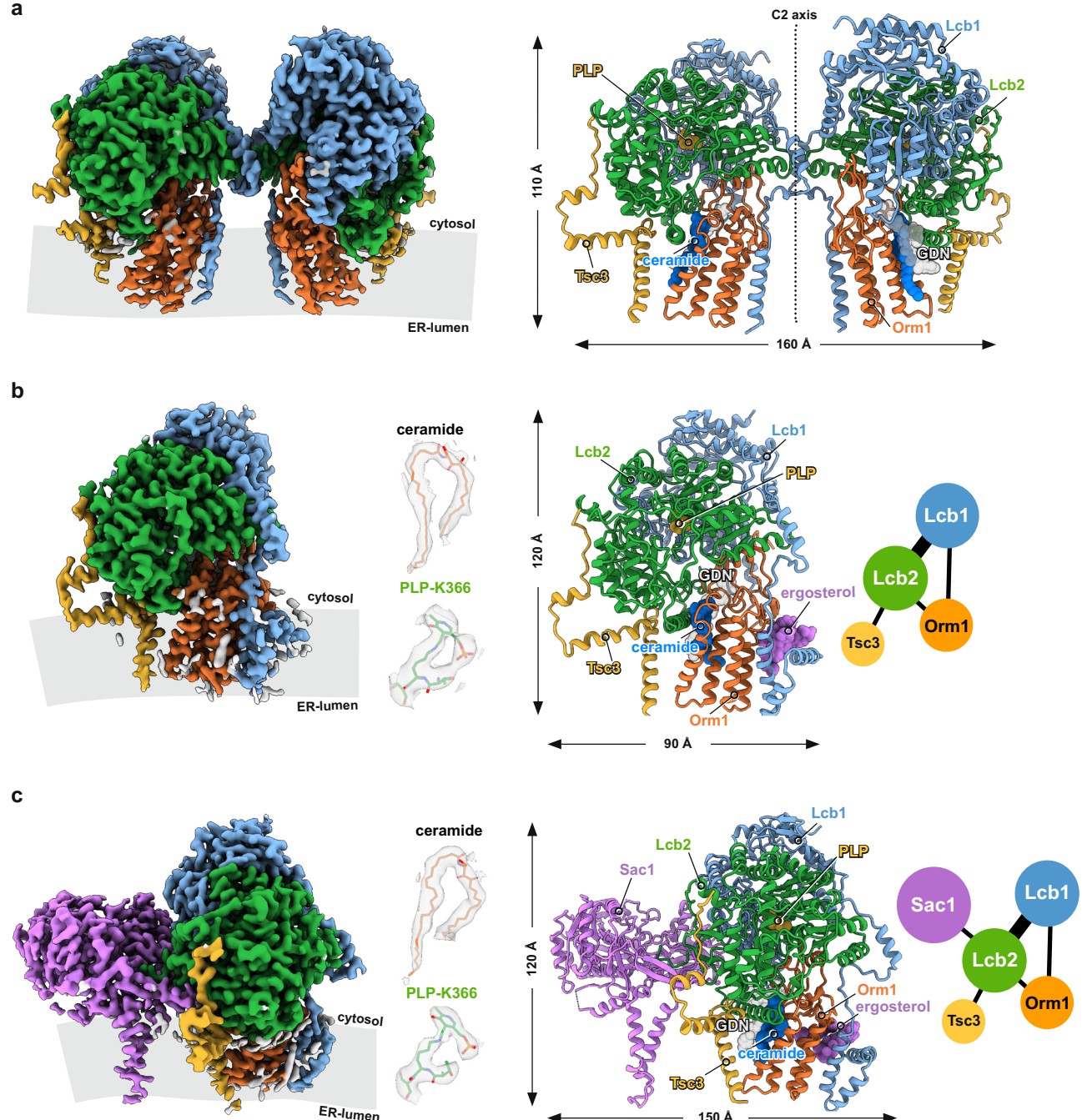

**Fig. 1 | Overall architecture of yeast SPT complexes. a** Cryo-EM density map and model of the C2 symmetric yeast SPOT dimer at 3.4 Å resolution. **b** The map and the model of the monomeric SPOT complex at 3.4 Å resolution. The middle inset shows the local densities for 44:0;4 ceramide and PLP-K366. The contact diagram of the subunits within the complex is displayed on the right. The thickness of each line in the contact diagrams is proportional to the contact area between subunits. **c** The monomeric SPOTS complex at 3.3 Å resolution. The contact diagram is shown on the right and the local densities for 44:0;4 ceramide and PLP-K366 are in the middle. All individual subunits are consistently colored: Lcb1 in blue, Lcb2 in green, Orm1 in orange, Tsc3 in yellow, and Sac1 in purple. The membrane plane is indicated in gray.

and targeted lipidomics. As a control, an Orm-free preparation of the SPT-Tsc3-Sac1 complex was used. This analysis revealed the typical yeast 44:0;4 ceramides enriched in the SPOTS preparation, supporting its presence in the purified complex (Sup. Fig. 1h). Ceramides, the downstream metabolites of the SPT-catalyzed reaction, have been reported to negatively regulate SPT activity[26]. Therefore, the positioning of the 44:0;4 ceramide headgroup in the immediate proximity to the conserved, substrate-tunnel-gating PATP loop and its direct interactions with Y485 and Y110 of Lcb2 (Fig. 3b) effectively

blocks access to the substrate tunnel from the membrane, highlighting a regulatory mechanism of ceramide-based inhibition of SPT activity. To test this hypothesis, we mutated either Y485 or Y110 to serine. Mutating Y485, which is only conserved in yeast and *D. discoideum* (Fig. 3f), yielded viable progenies in tetrad dissection (Fig. 3d) that had higher SPT activity as determined by measuring 3-KS, LCB, and ceramide levels in the cells (Fig. 3e). In contrast, replacing the highly conserved Y110 (Fig. 3f) to serine is lethal (Fig. 3d). This lethal phenotype could be rescued by supplementation

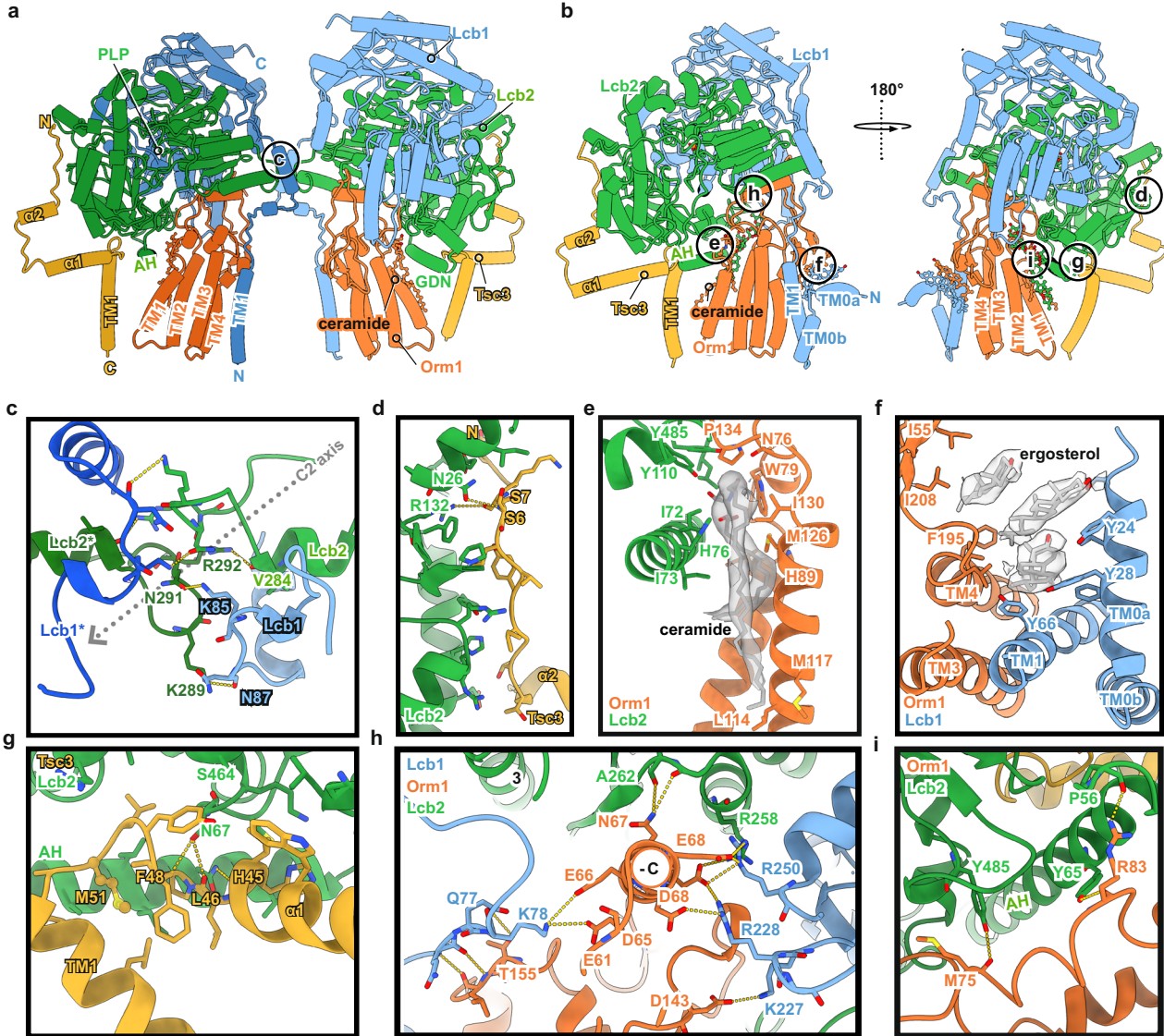

**Fig. 2 | Overview of the specific subunit interactions within the SPOT complex.** In the dimeric (**a**) and the monomeric (**b**) complexes, Orm1 interacts with Lcb1 and Lcb2. Lcb1 and Lcb2 interact tightly with each other; while Tsc3 binds exclusively to Lcb2. Color code is the same as in Fig.1. The Lcb1 subunit of the second protomer is shown in darker blue to provide a better view on the dimer interface. The circles indicate the areas of zoom-in views on the specific interactions in panels c-i. **c** Close-up view on the dimer interface, formed by the interactions between the cytosolic residues of Lcb1 and Lcb2. **d**, **g** Close-up views on the interactions between Tsc3 and Lcb2, **e** Ceramide 44:0;4 binding to Lcb2 and Orm1, **i** Interactions of the amphipathic helix (AH) of Lcb2 with Orm1, **h** Orm1 interactions with Lcb1 and Lcb2 via a short α-helix (-C = neg. electrostatic potential), **f** Ergosterol binding to Lcb1 and Orm1. Residues, which mediate key interactions are shown as sticks. Polar contacts are indicated with yellow dotted lines. The subunits are depicted as cartoons and ligands are shown in ball-and-stick representation. The experimental densities for ceramide and ergosterol are shown in transparent gray.

of phytosphingosine (PHS, Fig. 3d), suggesting that the mutation destabilizes the adjacent PATP loop and thus affects the enzymatic activity. Indeed, expressing the Y110S allele in a WT strain yields a dominant negative phenotype with lower sphingolipid levels measured (Fig. 3e).

In addition, we generated mutations in the Lcb2 amphipathic helix and the transmembrane domain 2 of Orm1. We exchanged the hydrophobic L69 in Lcb2 and either G122 or M126 in Orm1 with bulky phenylalanines to create a steric hindrance for the incorporation of ceramide into the complex. Both double mutants (Lcb2[L69F] Orm1[G122F] and Lcb2[L69F] and Orm1[M126F]) showed increased levels of 3-KS, LCBs, and ceramides (Fig. 3h) and grew better in the presence of myriocin compared to control cells (Fig. 3g). This suggests that the SPT in these strains is more active.

## The structure of the SPOT monomer

Previous studies were confined to dimeric SPT-complex preparations, and monomeric structures arise from focused classification and refinements. The crossover helices between the SPTLC1 subunits in the SPT-ORMDL3 structures offer a simple explanation of why the purification of individual monomeric complexes has not been possible previously. To analyze the predominant form of the SPOT complex in yeast cells, we generated diploid cells expressing one FLAG-tagged and one ALFA-tagged copy of Lcb1. In our pulldown experiments, both constructs barely co-purified with the differentially tagged Lcb1 subunit, suggesting that monomers are the predominant form of the SPOT complex in yeast cells (Sup. Fig. 1e).

Superposition of the monomeric SPOT complex, solved to 3.4 Å resolution (Fig. 1b), with the dimeric version reveals only minor overall

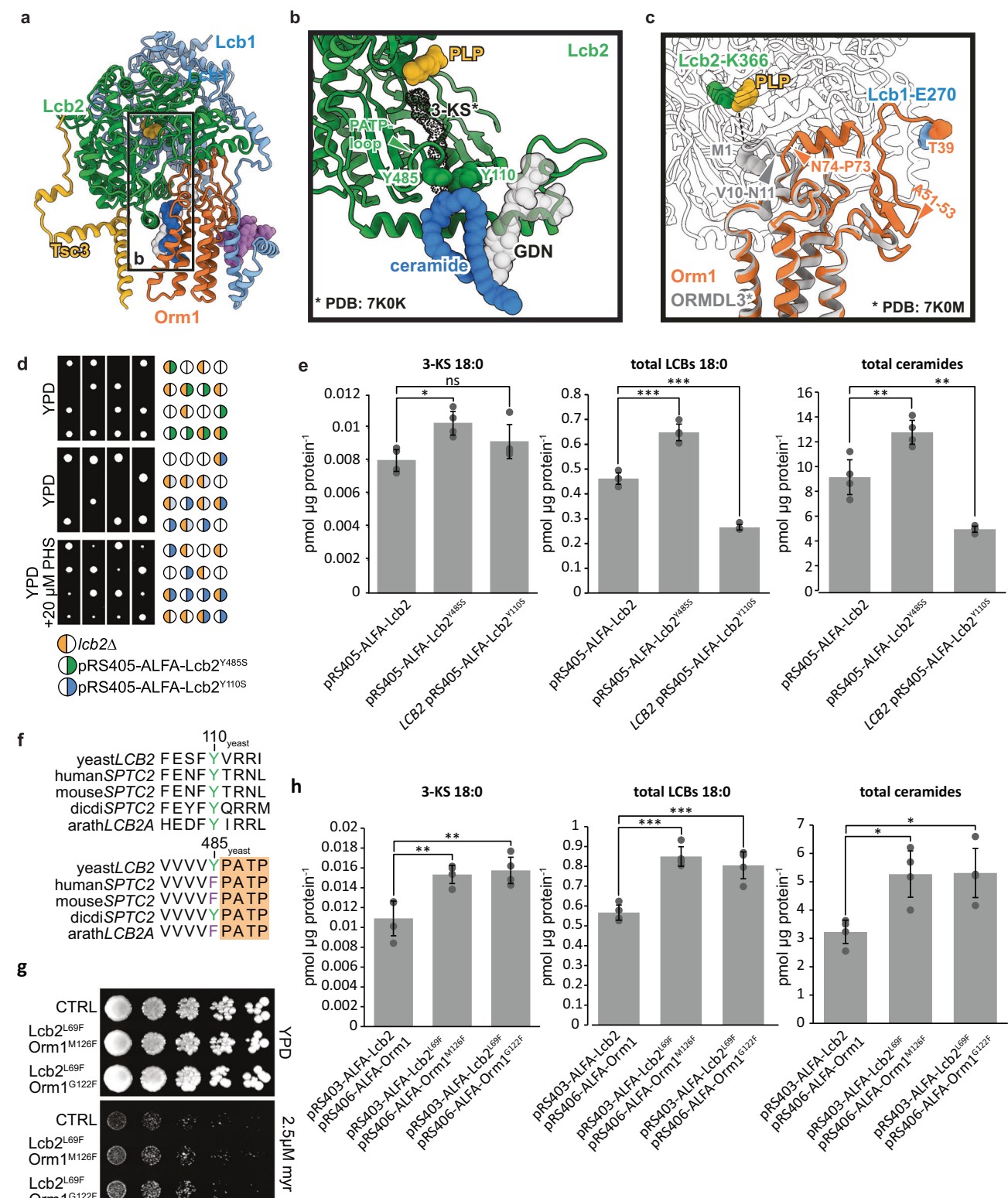

differences (RMSD = 0.48 Å across all 554 pairs, Sup. Fig. 7g). Most notably is an interrupted transmembrane helix at the N-terminus of Lcb1, that is absent in humans (Sup. Fig. 9d). In humans, the N-terminus of SPTLC1 starts with a short amphipathic helix located in the ER lumen, consecutively entering the membrane as the previously mentioned crossover helix. While the position of the corresponding transmembrane helix Lcb1-TM1 is virtually identical in yeast, the organization of the N-terminus is very different (Fig. 2b, f). Preceding the TM1, Lcb1 folds into a short transmembrane helix (TM0b) that

spans approximately half of the bilayer (15 Å in length) and leads through a short loop into another helix, that runs parallel to the membrane and is deeply embedded in the upper leaflet (TM0a). This places the Lcb1 N-terminus in the cytosol, which was confirmed by its ability to recruit a cytosolic GFP-tagged ALFA nanobody to the ER membrane (Sup. Fig. 10). The TM0a helix also forms a hydrophobic pocket with the Orm1-TM4, in which three structurally well-resolved ergosterol molecules are positioned (Fig. 2f, Sup. Fig. 5c). The N-terminal helices are not resolved in the dimeric SPOT complex, and

**Fig. 3 | Regulation of SPT activity by Orm1 and ceramides. a** Ligand binding within the SPOT complex. The color code is the same as in Fig. 1. Ligands are represented as spheres with ergosterol in violet, 44:0;4 ceramide in dark blue, PLP in yellow, and GDN in semi-transparent gray. **b** Blocking of the substrate access tunnel by the putative Lcb2-gatekeeper residues Y110 and Y485 and ceramide 44:0;4, which is further stabilized by GDN. A docked 3-KS molecule (PDB: 7K0K, dotted black density) indicates the upper region of the substrate access tunnel. **c** Superposition of Orm1 and ORMDL3 (PDB: 7K0M) highlights divergence of M1-ORMDL3 towards the active site at Lcb2-K366 and Orm1-T39 towards Lcb1-E270. Diverging residues of ORMDL3 (V10-N11) and Orm1 (N74-P73) are marked with a triangle. Phosphorylation-sites are indicated with an orange triangle (residues mutated from serine to alanine). Other subunits were omitted for clarity. **d** Tetrad analysis of *lcb2Δ cells* (yellow), expressing *Lcb2^Y48SS* (green; upper panel) or *Lcb2^Y110S* (blue) in the absence (middle panel) or presence of 20 μM PHS (lower panel). **e** Levels of 3-ketosphinganine (3-KS), long chain bases (LCBs), and ceramides in

*lcb2Δ* cells expressing ALFA-Lcb2 or ALFA-*Lcb2^Y48SS* and WT cells expressing ALFA-*Lcb2^Y110S*. Data were analyzed by one-way ANOVA with Tukey's multiple-comparison test (*$p < 0.05$, **$p < 0.01$, ***$p < 0.001$). Exact $P$-values are shown in Sup. Tab. 7. **f** Sequence alignments of the region surrounding the conserved Y residues in different species. **g** Serial dilutions of *lcb2Δ orm1Δ* cells expressing ALFA-Lcb2 and ALFA-Orm1, ALFA-*Lcb2^L69F* and ALFA-*Orm1^M126F* or ALFA-*Lcb2^L69F* and ALFA-*Orm1^G122F* on YPD plates (control) and YPD plates containing 2.5 μM myriocin. **h** Levels of 3-ketosphinganine (3-KS), long chain bases (LCBs) and ceramides in *lcb2Δ orm1Δ* cells expressing ALFA-Lcb2 and ALFA-Orm1 (left) or ALFA-*Lcb2^L69F* and ALFA-*Orm1^M126F* (middle) or ALFA-*Lcb2^L69F* and ALFA-*Orm1^G122F* (right). Data were analyzed by one-way ANOVA with Tukey's multiple-comparison test (*$p < 0.05$, **$p < 0.01$, ***$p < 0.001$). Exact $P$-values for e and h are shown in Sup. Tab. 7. $n = 4$ biologically independent samples for e and h and data are presented as mean values ± SD. Source data are provided as a Source Data file for d, e, g, and h.

superposition of two SPOT-monomer models onto each C2 symmetric protomer within the SPOT dimer model results in a sterical clash of adjacent Lcb1-TM0a (Sup. Fig. 7g). The presence of sterol molecules was confirmed by an enzyme-coupled reaction, resulting in the detection of approx. 15 ng ergosterol per μg protein from the purified complex (Sup. Fig. 1g). Deletion of either the TM0a or the entire TM0 still allowed purification of the complex (Sup. Fig. 12a). However, deletion of TM0a led to a minor reduction in SPT activity, determined by LCB and ceramide levels (Sup. Fig. 12b). The direct interaction of the SPT with ergosterol offers a potential explanation for the co-regulation of sterols and sphingolipids in yeast, as discussed before[6].

### Structure of the SPOTS complex
Previous works have identified the interaction of the PI4P phosphatase Sac1 with the SPOT complex; however, the role of Sac1 within the complex remains elusive. Here, we solved the structure of the SPOTS complex at 3.3 Å resolution (Fig. 1c). In the crystal structure of the Sac1 phosphatase domain, large parts of this domain have not been resolved and thus interpreted as flexible regions (PDB: 3LWT)[29]. In our cryo-EM structure, Sac1 tightly interacts with the flexible N-terminus of Lcb2 and is anchored to the lipid bilayer via two transmembrane helices and an amphipathic helix (Fig. 4a, Sup. Fig. 6). Membrane-embedded Sac1 within the SPOTS complex does not show significant flexibility. Sac1 interacts with Lcb2 exclusively, and we do not observe a previously suggested interaction with Tsc3[33]. The binding interface between Sac1 and Lcb2 involves the eight amino-terminal residues of Lcb2 and a C-terminal hairpin β-sheet motif in Sac1 (Fig. 4c, Sup. Fig. 9a). A smaller binding interface involves a H-bond between Sac1^I104 - Lcb2^N88 and several non-polar contacts (Fig. 4b). Interestingly, Sac1 is only bound to monomeric SPOT complexes in our data. To directly test the interaction between Sac1 and the SPOT complex, we deleted the C-terminal β-hairpin (*sac1Δ574*) of Sac1. We compared the interaction partners from ALFA-Sac1 pulldowns and ALFA-*sac1Δ574* pulldowns using mass spectrometry-based proteomics (Fig. 4e, f). As predicted, we significantly co-enriched all subunits in the WT pulldowns (Lcb1, Lcb2, Tsc3, Orm1, Orm2; Fig. 4e) and these interaction partners were lost in the pulldowns of the Sac1 mutant (Fig. 4f).

To investigate local sequence conservation within the Sac1-Lcb2-interface, homologs from human and *D. discoideum*, lacking the C-terminal hairpin β-sheet motif, were used for multiple sequence alignment, which revealed poor conservation of key-residues across species (Sup. Figs. 7f, 9a). However, a superposition of the AlphaFold model of *D. discoideum* Sac1 with our experimental structure suggests a conserved basis for Lcb2-binding independent of the Sac1 β-sheet motif (Sup. Fig. 7e, h).

To test if the SPOTS complex also exists in other organisms, we analyzed GFP-Sac1 pulldowns from *D. discoideum* (Fig. 4g) using mass spectrometry-based proteomics. These experiments revealed the presence of the SPOTS complex also in *D. discoideum*. *D. discoideum*

Sac1 showed interactions with the two SPT subunits, sptA and sptB, and the Orm1-like protein 2. The β-hairpin motif of *S. cerevisiae* Sac1 can thus be interpreted as a yeast-specific regulatory structure.

It was previously shown that the deletion of Sac1 affects the levels of sphingolipids in yeast. However, the canonical function of Sac1 is the dephosphorylation of PI4P, which is important to maintain sterol transport from the ER to the Golgi apparatus via the OSBP homolog Osh4[34]. The deletion of the β-hairpin motif of Sac1 now allowed us to uncouple the phosphatase activity of Sac1 from its function in the SPOTS complex. Our analysis of 3-KS, LCBs, and ceramides confirmed the increased levels caused by the deletion of *SAC1* compared to WT cells (Fig. 4d). Interestingly, the deletion of the Sac1 C-terminus also caused increased levels of LCBs and ceramides, but less dramatic than the deletion of the entire gene (Fig. 4d). This suggests that Sac1 negatively regulates the activity of the SPT via its interaction with Lcb2 independent of its role in sterol metabolism. Since we do not observe any conformational changes in the active site between the SPOT complex and the SPOTS complex, the exact molecular mechanism of SPT activity regulation by Sac1 remains to be further investigated.

## Discussion
The SPT enzyme is the rate-limiting factor in sphingolipid metabolism and controls cellular sphingolipid homeostasis through multiple input signals. Here, we present the structure of the yeast SPOT complex as both a monomer and a dimer and also the SPOTS complex, including the PI4P phosphatase Sac1. The SPOT complex is highly conserved across different species, as evidenced by the similarities in structure between yeast and mammalian SPT-Orm complexes. However, we also observe marked differences that explain their different regulatory mechanisms.

The lack of the Lcb1-TM1 helix swap, previously observed in human SPT complexes, results in weaker protomer interactions and explains the presence of monomeric complexes. Both oligomeric states from the human complexes are active in vitro; however, their respective physiological relevance is unclear. Importantly, we only detected the regulatory subunit Sac1 in interaction with the monomer. Co-purification of Orm1 and Orm2 is low when the complexes are purified through the Orm subunits[21]. It is also difficult to conceive that the cell is able to discriminate between the two highly homologous Orm proteins during the loading of a dimeric SPOT complex. Finally, Orm2 is exclusively regulated through the endosome/Golgi-associated degradation (EGAD) pathway[35]. Together with our interaction studies, this suggests that the monomeric SPOT and SPOTS complexes are the predominant forms in yeast.

The membrane-spanning helices of Lcb1 in our structures differ largely from the currently annotated membrane topology. Biochemical studies suggested the presence of three transmembrane helices, which are spread across far-apart regions of Lcb1 (amino acids 50–84, 342–371, 425–457)[13]. In all of our structures, the yeast Lcb1 TM1 is

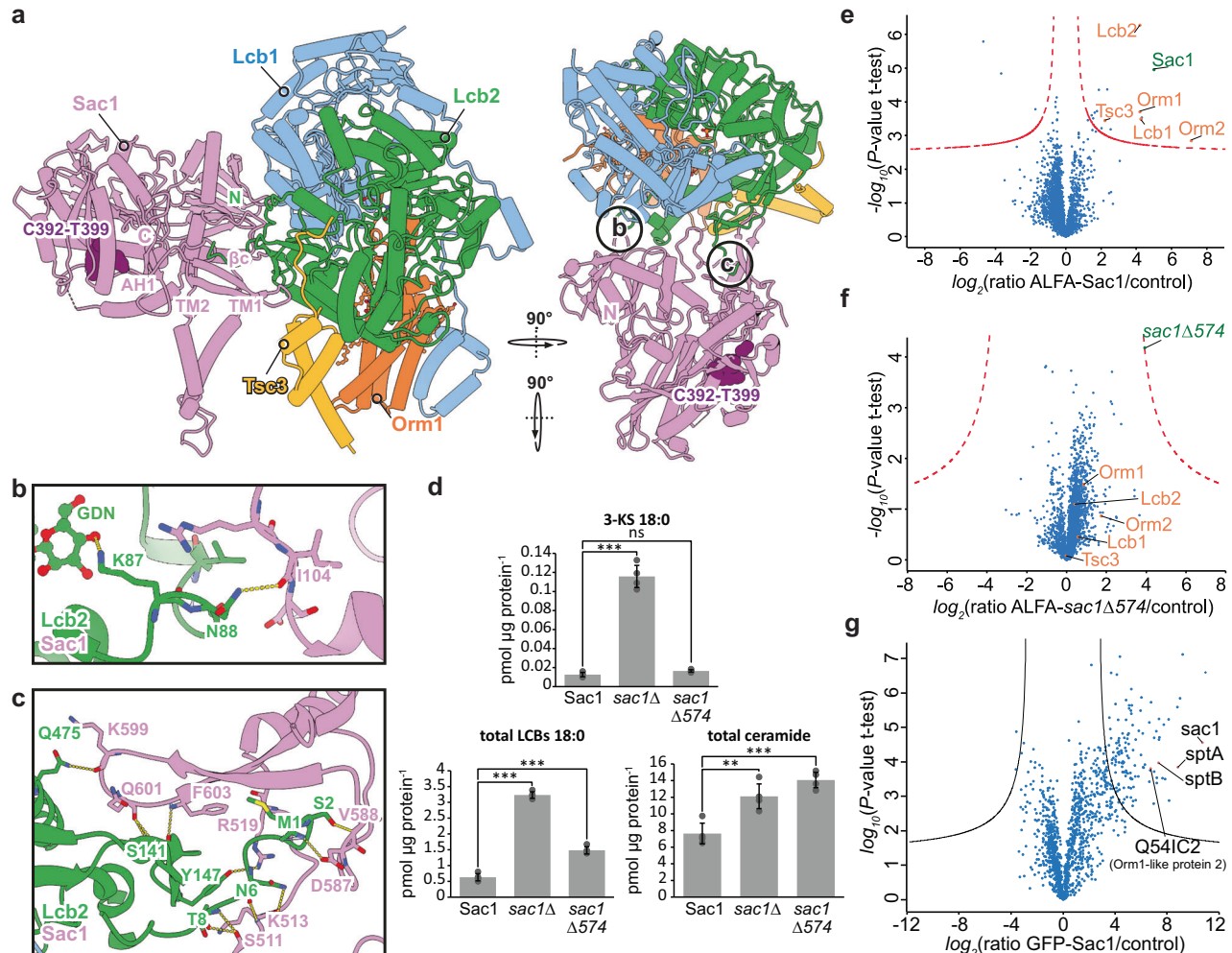

**Fig. 4 | Sac1 binding to the SPOT complex. a** Sac1 interactions within the SPOTS complex. Color code is the same as in Fig. 1. **b, c** Close-up views of the interactions between Sac1 and Lcb2. Polar contacts are indicated with yellow dotted lines. The subunits are depicted as cartoons and ligands are shown in ball-and-stick representation. **d** Levels of 3-ketosphinganine (3-KS), long chain bases (LCBs), and ceramides in *sac1Δ* cells harboring a plasmid expressing ALFA-Sac1, *sac1Δ* cells or *sac1Δ* cells harboring a plasmid expressing ALFA-*sac1Δ574*. Data were analyzed by one-way ANOVA with Tukey's multiple-comparison test (*$p < 0.05$, **$p < 0.01$, ***$p < 0.001$) with n = 4 biologically independent samples and data are presented as mean values ± SD. Exact *P*-values are shown in Sup. Tab. 7. **e** Label free proteomics of yeast cells expressing ALFA-Sac1 compared to untagged control cells. In the volcano plot, the protein abundance ratios of ALFA-Sac1 over control cells are plotted against the negative $log_{10}$ of the *P*-value of the two-tailed t-test for each protein. **f** Label free proteomics of yeast cells expressing ALFA-*sac1Δ574* compared to untagged control cells. In the volcano plot, the protein abundance ratios of ALFA-*sac1Δ574* over control cells are plotted against the negative $log_{10}$ of the *P*-value of the two-tailed t-test for each protein. **g** Label free proteomics of *D. discoideum* cells expressing GFP-Sac1 compared to untagged control cells. In the volcano plot, the protein abundance ratios of GFP-Sac1 over control cells are plotted against the negative $log_{10}$ of the *P*-value of the two-tailed t-test for each protein. Source data are provided as a Source Data file for **d**, **e**, and **f**.

located at the same position as the S1 helix of human SPTLC1. Additionally, we also find one more interrupted N-terminal membrane inserted helix (TM0 with TM0a: T20-Q35 and TM0b: Q40-S49) in front of the TM1 helix in the monomeric species.

Despite the high structural resemblance, the Orm-dependent SPT regulation is different in yeast and humans. In human cells, the N-terminus of ORMDL3 regulates access to the substrate binding pocket of SPTLC2[19]. In yeast, both Orm1 termini face away from the active site of Lcb2, requiring another mechanism of regulation. Additionally, the phosphorylation of three serine residues on the N-terminal loop of Orm1/2 has been discussed to influence SPT activity[23–25]. Notably, these amino acids are located at a highly accessible position, connecting the N-terminus of Orm1 to Lcb1. Therefore, it can be anticipated that phosphorylation at this position leads to the rearrangement of the complex, potentially affecting its stability or activity. Here, we analyzed the structure of the Orm1 containing SPOTS complex. Orm1 and Orm2 are highly conserved on the sequence level,

yet their deletion affects the activity of the SPT differentially[15,36]. It remains possible that the different effect of Orm1 and Orm2 regulation depends on the cellular localization of both complexes or the accessibility by the Ypk kinases[36].

Furthermore, our data explain the ceramide-induced regulation that has been reported for yeast SPT complexes, which is dependent on Orm binding[26]. In all of our structures, TM1-2 of Orm1 acts as a docking station for a 44:0;4 ceramide, which is sandwiched between Orm1 and the amphipathic helix of Lcb2. The ceramide extends out of the cytosol-facing membrane plane through interactions with two aromatic residues, Y110 and Y485, of Lcb2. We propose that the two tyrosine residues function as gatekeepers controlling access to the previously discovered substrate channel[19]. Mutating the gatekeeper residues or introducing bulky hydrophobic amino acids in the Lcb2 amphipathic helix and the Orm1 TM2 directly affects SPT activity. We also observe a decrease in ceramide levels in Orm1-free SPT samples, further supporting the Orm1-mediated regulatory effect of ceramide.

Two recent studies also present models for the ceramide-mediated inhibition of the SPT-Orm complex in both, mammalian cells and plant cells[37,38]. Interestingly, in mammalian cells, the interaction of ceramide with an asparagine residue locks the N-terminus of ORMDL3 in the active site. In plant cells, the presence of ceramide leads to the formation of a hybrid β-sheet between Lcb2 and Orm1 to lock the N-terminus (Sup. Fig. 7d). Together, this suggests that the ceramide-mediated inhibition of the SPT is evolutionary conserved but the molecular mechanism differs between species. Notably, only our structure harbors an endogenous species-specific ceramide.

It is an interesting observation that ceramide is not the only co-purified lipid in the monomeric SPOT and SPOTS complexes. The deeply embedded helix Lcb1-TM0a directly interacts with three molecules of ergosterol in a position that would correspond to the upper leaflet of the ER membrane. Levels of ergosterol and sphingolipids have been previously suggested to be tightly connected to each other[6,7,39]. It is appealing to speculate that SPT activity could be directly regulated via the levels of ER-localized ergosterol. Since the superposition of two monomeric structures onto a dimeric complex causes steric clashes of TM0a helices (Sup. Fig. 7g), it is possible that high levels of ergosterol promote the formation of the TM0a helix, breaking the dimer apart and thus changing SPT activity.

Sac1 has been shown to interact with the SPOT complex and was proposed to affect its activity[21]. The canonical function of Sac1 is the dephosphorylation of PI4P in the ER. PI4P is generated at the Golgi apparatus and is exchanged with ER-synthesized sterols via the oxysterol-binding proteins (OSBPs)[40–42]. This exchange is driven by the Sac1-dependent dephosphorylation of PI4P in the ER. We show that Sac1 specifically interacts with the N-terminus of the Lcb2 subunit and not with Tsc3, as suggested previously[33]. A short C-terminal β-hairpin structure supports the interaction. Interestingly, uncoupling the phosphatase activity of Sac1 from its interaction with the SPOT complex affects SPT activity, showing that the interaction has a regulatory function. The structure of Sac1 includes two transmembrane helices and an additional amphipathic helix. This shows that the previously uncharacterized region between the phosphatase domain and the transmembrane helices is structured when in contact with a hydrophobic moiety, proving that Sac1 can only act at the membrane where it is inserted[29,43–46].

In summary, we present a detailed picture of the protein interactions within the SPOTS complex, which is at the heart of neurological disorders such as HSAN1 and childhood ALS[9,47–49]. We reveal how SPT activity in yeast is controlled through its downstream metabolite ceramide. The additional ergosterol binding site provides a mechanistic link in the co-regulation of sphingolipid and sterol metabolism.

## Methods

### Yeast strains
Yeast strains and plasmids used in this study are listed in Supplementary Tables 2 and 4. For purifications, SPOTS subunits were expressed under the control of the *GAL1* promoter using integrative plasmids. The 3xFLAG tag was inserted between codons 9 and 10 of *LCB1*, as previously reported[21].

### Purification of 3xFLAG tagged SPOTS complex from *S. cerevisiae*
Yeast cells were collected after growth for 24 h in yeast peptone (YP) medium containing 2% galactose (v/v), washed in lysis buffer (50 mM HEPES-KOH (pH 6.8), 150 mM KOAc, 2 mM MgOAc, 1 mM CaCl$_2$, 200 mM sorbitol) and resuspended in a 1:1 ratio (w/v) in lysis buffer supplemented with 1 mM phenylmethylsulfonylfluoride (PMSF) and 1x FY protease inhibitor mix (Serva). Resuspended cells were frozen in a drop-by-drop fashion in liquid nitrogen, pulverized in 15 × 2 min cycles at 12 CPS in a 6875D Freezer/Mill Dual-Chamber Cryogenic Grinder (SPEX SamplePrep), and thawed in lysis buffer with 1 mM PMSF and 1x FY. After two centrifugation steps at 1000 $g$ at 4 °C for 20 min,

microsomal membranes were pelleted at 44,000 $g$ at 4 °C for 30 min. Cells were resuspended in lysis buffer and then diluted with IP buffer (50 mM HEPES-KOH, pH 6.8, 150 mM KOAc, 2 mM MgOAc, 1 mM CaCl$_2$, 15% glycerol) with 1% glyco-diosgenin (GDN) supplemented with protease inhibitors. After nutating for 1.5 h at 4 °C, unsolubilized membranes were pelleted at 44,000 $g$ at 4 °C for 30 min. The supernatant was added to α-FLAG resin (Sigma Aldrich) and nutated for 45 min at 4 °C. Beads were washed twice with 20 ml IP buffer with 0.1% GDN and 0.01% GDN, respectively. Bound proteins were eluted twice on a turning wheel in IP buffer with 0.01% GDN for 45 min and 5 min, respectively, at 4 °C with 3xFLAG peptide. The eluates were collected by centrifugation at 460 $g$ at 4 °C for 30 s and concentrated in a 100 kDa Amicon Ultra centrifugal filter (Merck Millipore), which was equilibrated with IP buffer containing 1% GDN. The concentrated eluate was applied to a Superose 6 Increase 5/150 column (Cytiva) for size exclusion chromatography (SEC) and eluted in 50 µl fractions using ÄKTA go purification system (Cytiva). Peak fractions were collected, concentrated as described before, and used for further analysis.

### Cryo-EM sample preparation and data acquisition
Sample quality was inspected by negative-stain electron microscopy as previously described[50]. Micrographs of the negatively-stained sample were recorded manually on a JEM2100plus transmission electron microscope (Jeol), operating at 200 kV and equipped with a Xarosa CMOS (Emsis) camera at a nominal magnification of 30,000, corresponding to a pixel size of 3.12 Å per pixel.

For cryo-EM, the sample was concentrated to 10 mg/ml. C-flat grids (Protochips; CF-1.2/1.3-3Cu-50) were glow-discharged, using a PELCO easiGlow device at 15 mA for 45 s and 3 µl of the concentrated sample were immediately applied and plunge frozen in liquid ethane, using a Vitrobot Mark IV (Thermo Fisher) at 100% relative humidity, 4 °C. The dataset was collected using a Glacios microscope (Thermo Fisher), operating at 200 kV and equipped with a Selectris energy filter (Thermo Fisher) with a slit of 10 eV. Movies were recorded with a Falcon 4 direct electron detector (Thermo Fisher) at a nominal magnification of 130,000 corresponding to a calibrated pixel size of 0.924 Å per pixel, and the data was saved in the electron-event representation (EER) format. The dose rate was set to 5.22 e$^-$ per pixel per second and a total dose of 50 e$^-$ per Å$^2$. 13,604 movies were collected automatically, using EPU software (v.2.9, Thermo Fisher) with a defocus range of −0.8 to −2.0 µm.

### Cryo-EM image processing
The SPOTS dataset was processed in cryoSPARC (v.4), and the processing workflow is presented in Sup. Fig. 2. Movies were preprocessed with patch-based motion correction, patch-based CTF estimation and filtered by the CTF fit estimates using a cut-off at 5 Å in cryoSPARC live (v.4), resulting in a remaining stack of 12,552 micrographs (representative micrograph given in Sup. Fig. 7c).

Well-defined 2D classes were selected and used for subsequent rounds of template-based 2D classification. 1,006,434 particles were extracted in a box of 432 pixels, and Fourier cropped to 216 pixels. Previously generated ab initio 3D reconstructions from live processing were used for several rounds of heterogeneous refinement, which resulted in two distinct, well-defined reconstructions. Both reconstructions were processed separately.

**SPOT-dimer-complex.** Classes corresponding to the newly termed SPOT-dimer-complex were subjected to non-uniform refinement, and 3D-aligned particles were re-extracted without binning. Additional rounds of heterogeneous refinement following non-uniform refinement and local refinement with C2 symmetry applied resulted in a stack of 142 K particles and a consensus map with 3.4 Å resolution.

To further improve the map quality, particles were symmetry expanded by using a C2 point group, thereby aligning signals coming

from both protomers onto the same reference map. Subsequent signal subtraction and local refinement results were subjected to 3D classification in PCA mode focused around the protomer. Additional local CTF correction and local refinement of 94,884 symmetry-expanded particles yielded a focused map with an overall resolution of 3.0 Å.

**SPOTS-monomer.** Similarly, a stack of 252,688 particles was re-extracted without binning using the alignment shifts from a heterogeneous refinement. Aligned particles were further classified through heterogeneous refinement and 3D classification in PCA mode. A final round of non-uniform-refinement with higher-order aberration correction enabled, local CTF correction, and local refinement resulted in an overall resolution of 3.3 Å.

**SPOT-monomer.** An additional well-resolved 3D class was identified, lacking the Sac1-assigned density, and processed separately. 123 K particles were cleaned through 2D classification to remove the remaining non-protomer classes. A stack of 96 K particles was subjected to heterogeneous refinement, and the remaining 89 K particles were further refined through non-uniform and local refinement. This resulted in a final map with a global resolution of 3.4 Å.

All maps were subjected to unsupervised B-factor sharpening within cryoSPARC. Reported B-factors resulted from un-supervised auto-sharpening during refinement in cryoSPARC. To aid model building, unsharpened half-maps were subjected to density modification within Phenix *phenix.resolve_cryo_em*.

### Model building and refinement
Initial atomic models were generated using the AlphaFold2 prediction of a monomer, which was placed in the individual maps and fitted as rigid bodies through UCSF Chimera[51]. The structure was manually inspected in Coot (v.0.9)[52] and iteratively refined using *phenix.real_space_refine* within Phenix (v.1.19). Calculation of compositional heterogeneity of the reconstructions was performed with "OccuPy"[32]. Validation reports were automatically generated by MolProbity[53] within Phenix[54]. All density maps and models have been deposited in the Electron Microscopy Data Bank and the PDB. The PDB IDs are 8C82 (SPOTS-Dimer-Complex), 8C80 (SPOTS-Orm1-Monomer) and 8C81 (SPOTS-Orm1-Monomer-Sac1). The respective EMDB IDs are EMD-16469, EMD-16467, and EMD-16468. All structural data was visualized with ChimeraX[51] and protein interactions were analyzed with the help of PDBe PISA[55]. 2D ligand-protein interaction diagrams were calculated in LigPlot+[56]. Characterization of selected helices was performed in HeliQuest[57]. An overview of the local density fit is given in Sup. Fig. 4.

### Pulldown experiments
For pulldown experiments, cells were inoculated from an overnight pre-culture in 200 ml YPD in independent biological triplicates ($n = 3$) and grown to an exponential growth phase at 30 °C. The same amounts of cells were harvested at 2272 $g$ at 4 °C for 5 min and snap frozen in liquid nitrogen. Cells were lysed with 500 μl glass beads in 500 μl pulldown buffer (20 mM HEPES pH 7.4, 150 mM KOAc, 5% glycerol, 1% GDN, Roche Complete Protease Inhibitor Cocktail EDTA free, Roche) using a FastPrep (MP biomedicals). After centrifugation at 1000 $g$ at 4 °C for 10 min, supernatants were incubated for 30 min at 4 °C on a turning wheel. Supernatants were spun down at 21,000 $g$ at 4 °C for 10 min and incubated on a turning wheel at 4 °C with 12.5 μl pre-equilibrated ALFA beads (NanoTag Biotechnologies). Beads were washed two times with pulldown buffer and then four times with wash buffer (20 mM HEPES pH 7.4, 150 mM KOAc, 5% glycerol). Proteins on beads were digested and further treated according to the iST Sample Preparation Kit (PreOmics) protocol. Dried peptides were resuspended in 10 μl LC-Load, and 2 μl were used to perform reversed-phase chromatography on a Thermo UltiMate 3000 RSLCnano system connected to a TimsTOF HT mass spectrometer (Bruker Corporation,

Bremen) through a Captive Spray Ion source. Peptides were separated on an Aurora Gen3 C18 column (25 cm x 75um x 1.6um) with CSI emitter (Ionoptics, Australia) at 40 °C. Elution of peptides from the column was realized via a linear gradient of acetonitrile from 10–35% in 0.1% formic acid for 44 min at a constant flow rate of 300 nl/min following a 7 min increase to 50%, and finally, 4 min to reach 85% buffer B. Eluted peptides were directly electro sprayed into the mass spectrometer at an electrospray voltage of 1.5 kV and 3 l/min Dry Gas. The MS settings of the TimsTOF were adjusted to positive ion polarity with a MS range from 100 to 1700 m/z. The scan mode was set to PASEF. The ion mobility was ramped from 0.7 Vs/cm2 to 1.5 in 100 ms. The accumulation time was set to 100 ms. 10 PASEF ramps per cycle resulted in a duty cycle time of 1.17 s. The target intensity was adjusted to 14000 and the intensity threshold to 1200. The dynamic exclusion time was set to 0.4 min to avoid repeated scanning of the precursor ions, their charge state was limited from 0 to 5. The resulting data were streamed to the PASer (Parallel Search Engine in Real-Time version 2023b from Bruker) and analyzed with ProLuCID database search algorithm with the corresponding FASTA databases. Precursors ranged from 600 to 6000 Da. Carbamidomethylation (C) and oxidation (M) were chosen as modifications. DDA-MBR were performed with MS tolerance of 10 ppm and IM tolerance of 0.05 (1/k0). Resulting data were analyzed using Perseus (V2.0.7.0, www.maxquant.org/perseus)[58]. Significance lines in the volcano plot of the Perseus software package corresponding to a given FDR were determined by a permutation-based method[59]. The mass spectrometry proteomics data have been deposited to the ProteomeXchange Consortium via the PRIDE[60] partner repository.

### Western blots
For western blot analysis of pulldown experiments, cells were inoculated from a pre-culture in 500 ml YPD and grown to exponential growth phase at 30 °C. Pulldown experiments were performed as described before with either 12.5 μl pre-equilibrated ALFA beads (NanoTag Biotechnologies) or Fab-Trap beads (ChromoTek). Samples and beads were boiled at 60 °C for 10 min in Laemmli buffer with DTT. FLAG-tagged proteins were detected with a mouse anti-FLAG (F1804, Sigma) antibody diluted 1:1,000 followed by an 1:10,000 diluted anti-mouse IgG secondary antibody conjugated to horseradish peroxidase (HRP, Thermo Scientific, 31430). ALFA-tagged proteins were detected using an 1:1,000 diluted rabbit anti-ALFA antibody (N1581, Nanotag) followed by a 1:20,000 diluted mouse anti-rabbit secondary antibody conjugated to HRP (Santa Cruz Biotechnology).

### LCB and ceramide analysis of purified SPOTS complex and whole cell lysate
LCBs and ceramide were extracted and measured from 20 μg of purified proteins in at least technical triplicates ($n = 3$) or from an equivalent of 150 μg protein from whole cell lysate in biological independent quadruplicates ($n = 4$). For the LC-MS/MS analysis of LCB and ceramides, cells were grown in YPD to exponential growth phase. 150 mM ammonium formate was added to the purified complex or cell lysate. As an internal standard, ceramide (Sphingosine d17:1, CER d17:1/24:0; Avanti) was added, and lipid extraction with 2:1 chloroform/methanol was performed as described previously[61,62]. Dried lipid films were dissolved in a 65:35 (v/v) mixture of Buffer A (50:50 water/acetonitrile, 10 mM ammonium formate, and 0.1% formic acid) and B (88:10:2 2-propanol/acetonitrile/water, 2 mM ammonium formate and 0.02% formic acid). An external standard curve was prepared using dihydrosphingosine 18:0 (DHS; Avanti Polar Lipids), phytosphingosine 18:0 (PHS; Avanti Polar Lipids) and phytoceramide t18:0/24:0 (Avanti Polar 567 Lipids/Cayman). Samples were analyzed on an Accucore C30 LC column (150 mm × 2.1 mm 2.6 μm Solid Core; Thermo Fisher Scientific) connected to a Shimadzu Nexera HPLC system and a QTRAP 5500 LC-MS/MS (SCIEX) mass spectrometer. For the gradient, 40% B was used for 0.1 min. The concentration of Buffer B was increased to

50% over 1.4 min. Followed by an increase to 100% over 1.5 min. 100% B was kept for 1 min and decreased to 40% B for 0.1 min. 40% B was kept until the end of the gradient. A constant flow rate of 0.4 ml/min was used with a total analysis time of 6 min and an injection volume of 1 or 2 µl. The MS data were measured in positive ion, scheduled MRM mode without detection windows (Sup. Tab. 6). For peak integration, the SciexOS software was used. The concentrations of all lipid species were calculated using the external standard curve. The lipid concentrations were expressed in pmol/µg protein.

### GFP-Trap pulldown from *Dictyostelium discoideum*

*Dictyostelium discoideum* strains (Sup. Tab. 3) expressing either pDM317-GFP-Sac1 or pDM317-GFP[63] were grown at 22 °C in HL5-C medium (ForMedium) supplemented with geneticin (G418, 5 µg/ml). Electroporation of *D. discoideum* was performed according to Paschke et al. 2018 with modification[64]. The cell number was determined (Countess II F2, Thermo Fisher Invitrogen), and $3 \cdot 10^7$ cells were used for each sample. Pulldowns were performed in biological independent triplicates (n = 3). Cells were pelleted and washed once in cold Soerensen-Sorbitol. Cell pellets were snap frozen in liquid nitrogen and lysed with glass beads in 500 µl GFP pulldown buffer (20 mM HEPES pH 7.4, 150 mM KOAc, 5% glycerol, 1% GDN, Roche Complete Protease Inhibitor Cocktail EDTA free, Roche) using a FastPrep (MP biomedicals). The supernatant was cleared at 21,000 g for 10 min and incubated for 10 min rotating at 4 °C together with 12.5 µl pre-equilibrated GFP-Trap beads (Chromotek). Beads were washed four times with GFP pulldown buffer at 2500 g for 2 min at 4 °C. Afterwards, they were washed two times with wash buffer (20 mM HEPES pH 7.4, 150 mM KOAc, 5% glycerol) at 2500 g for 2 min at 4 °C. Beads were further treated following the iST Sample Preparation Kit (PreOmics) protocol. Dried peptides were resuspended in 10 µl LC-Load, and 4 µl were used to perform reversed-phase chromatography as described above. Data were analyzed using MaxQuant (V2.2.0.0, https://www.maxquant.org/maxquant/)[65,66] with the corresponding FASTA database. Contaminants were identified based on the MaxQuant contaminants.fasta file. The resulting data were analyzed using Perseus (V2.0.7.0, www.maxquant.org/perseus)[58]. Significance lines in the volcano plot of the Perseus software package corresponding to a given FDR were determined by a permutation-based method.[59] The mass spectrometry proteomics data have been deposited to the ProteomeXchange Consortium via the PRIDE[60] partner repository.

### Colorimetric-based enzymatic assay

SPT activity was measured by monitoring the release of CoA-SH from the SPT-catalyzed condensation of palmitoyl-CoA and L-serine. All assays were performed on a 200-µl scale. Briefly, all assays were performed in IP buffer with 0.008% GDN, 15 mM L-serine, 100 µM palmitoyl-CoA, and 30 µM PLP. The reaction was initiated by adding 1 µg protein. To validate SPT activity, 100 µM myriocin was added to inhibit SPT activity. Since myriocin was dissolved in methanol (MeOH), an appropriate amount of MeOH was added to all other samples. After incubation at RT for 1 h, the samples were deproteinized with a 3 kDa MWCO concentrator (Merck Millipore). To measure CoA levels, the samples were further treated in a 96-well plate following the Coenzyme A Assay Kit protocol from Sigma-Aldrich. The absorbance of the colorimetric product (570 nm) was measured in a SpectraMax iD3 Multi-Mode microplate reader. Corrected absorbances were obtained by subtracting the absorbance of the protein-free samples from the absorbances of the protein-containing samples.

### Fluorescence-based ergosterol measurements from purified protein

Bound ergosterol levels were measured using an enzymatic coupled assay generating the highly fluorescent dye resorufin. All assays were performed on a 100-µl scale in a 96-well plate according to the Amplex™ Red Cholesterol Assay Kit protocol (ThermoFisher Scientific). No cholesterol esterase was added. Fluorescence was recorded ($\lambda_{EX}$ = 550 nm, $\lambda_{EM}$ = 590 nm) after three hours using the SpectraMax iD3 Multi-Mode microplate reader. Relative fluorescence intensity was obtained by subtracting the fluorescence intensity of the protein-free samples from the intensity of the protein-containing samples.

### Fluorescence microscopy

Cells were grown overnight at 30 °C in a synthetic medium supplemented with essential amino acids (SDC), diluted in the morning to an $OD_{600}$ of 0.15, and grown to a logarithmic growth phase. Cells were directly imaged live in SDC medium using a Zeiss Axioscope 5 FL (Zeiss) equipped with a Plan-Apochromat 100x (1.4 numerical aperture (NA) oil immersion objective and an Axiocam 702 mono camera. Data were acquired with ZEN 3.1 pro software and processed with ImageJ 2.1.0. (National Institutes of Health, Bethesda, MD; RRID: SCR_003070). Single medial planes of yeast cells are shown.

### Spotting assays

For spotting assays, cells were inoculated from an overnight pre-culture and grown to an exponential growth phase in YPD. They were serially diluted and spotted onto YPD plates with and without indicated myriocin concentrations. Plates were incubated at 30 °C for 2 days.

### Tetrad dissection

To perform tetrad analysis, diploid yeast cells were collected from an overnight culture by centrifugation and placed on 1% KOAc agar for sporulation at 30 °C. After 3 days, cells were suspended in 100 µl sterile water and incubated for 9 min at room temperature with 5 µl zymolyase 20 T (10 mg/mL; MP Biomedicals, Eschwege, Germany). 10 µl of the suspension were added to a YPD plate or a YPD plate containing 20 µM PHS, and spores were segregated using a Singer MSM400 micromanipulator (Singer 445 Instruments, Somerset, UK). Plates were incubated at 30 °C for 2 days and then stamped onto the respective selection plates.

## Data availability

All density maps and models have been deposited in the EMDB and the PDB. The PDB IDs are 8C82 [https://doi.org/10.2210/pdb8C82/pdb] (SPOTS-Dimer-Complex), 8C80 [https://doi.org/10.2210/pdb8C80/pdb] (SPOTS-Orm1-Monomer) and 8C81 [https://doi.org/10.2210/pdb8C81/pdb] (SPOTS-Orm1-Monomer-Sac1). The respective EMDB IDs are EMD-16469, EMD-16467, and EMD-16468. The mass spectrometry proteomics data have been deposited to the ProteomeXchange Consortium via the PRIDE partner repository PXD044586 (Interaction proteomics of yeast ALFA-Sac1) and PDX039848 (Interaction proteomics of *Dictyostelium discoideum*). Source data are provided with this paper.

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

## Acknowledgements

We thank Jürgen Heinisch (Osnabrück) for providing the GFP-tagged ALFA-nanobody plasmid. We thank Oliver Schmidt (Innsbruck) for providing the original 3xFLAG-Lcb1 plasmid. We thank Caroline Barisch (Osnabrück) for sharing her expertise in *Dictyostelium discoideum* cell culture and the Hilbi laboratory (Zürich) for the pDM317 plasmids expressing GFP-Sac1 and GFP. This work was funded by the DFG (SFB944 P20 to FF and P27 to AM, the SFB1557 P6 to FF, P11 to AM, FR 3647/2-2 and FR 3647/4-1 to FF, INST.190/196-1 FUGG and BMBF 01ED2010 to AM. JHS is supported by a fellowship from the Friedrich Ebert foundation.

## Author contributions

Investigation: J.H.S., C.K., S.L., B.E., and S.W. Formal analysis: J.H.S., C.K., B.E., S.L., S.W., and K.P. Visualization: J.H.S., C.K., K.P., and F.F. Conceptualization: F.F., A.M., and D.J. Funding acquisition: F.F., A.M. Writing—original draft: J.H.S., C.K., F.F., A.M., and D.J. Writing—review and editing: J.H.S., C.K., F.F., A.M., and D.J.

## Funding

## Competing interests

The authors declare no competing interests.
