## [Peer Review File · Nature Communications]

REVIEWER COMMENTS

Reviewer #1 (Remarks to the Author):

Ceramides are lipids that play a variety of vital roles in cellular homeostasis. Dysregulated synthesis of these sphingolipids can lead to catastrophic consequences and is associated with a variety of pathologies. Activity of the serine palmitoyltransferase (SPT) enzyme, which is responsible for initiating the rate-limiting step in sphingolipid biosynthesis, is modulated by its association with ORM/ORMDL proteins and by cellular levels of ceramides and sterols. SPT is highly conserved across species, and in humans it is a dimer that further associates with the regulatory partners ORMDL3 and ssSPTa. Activity of SPT enzymes is further regulated by multiple cellular factors, such as levels of sphingolipids or sterols, and phosphorylation. The structural bases for these diverse modes of regulation remain poorly understood. Here, the authors report the structures of the SPT-Orm-Tsc3 complex (SPOT) in dimer and monomer forms, as well as of that of SPOT in complex with PI4P phosphatase Sac1 (SPOTS complex). These structures show that in the yeast complex the dimer interface is reduced, which allows the SPOT complex to also exist in monomeric form. Further, they show that Sac1 only binds to the SPOT monomer and the structures show how its association is sterically incompatible with formation of a SPOT dimer. Finally, these structures reveal the presence of several non-protein densities associated with the complex, which the authors assign to 44:0;4 ceramide and ergosterol molecules respectively. Notably, the 44:0;4 ceramide density is coordinated between the Orm1 and Lcb1 subunits and blocks the SPT substrate channel, suggesting a novel regulatory mechanism of SPT activity by sphingolipids.

Overall, the work is interesting, and the structures reported here provide insights into the assembly and regulation of this important enzyme. However, the lack of functional validation of the structure-driven inferences limits the interest of the present manuscript. The authors take great care of placing their structures in context of past results and illustrate how the present work helps in rationalizing these data. However, all insights obtained from their work remain hypothetical and speculative. The present manuscript would be greatly strengthened by functional work where specific prediction are tested.

For example, the authors claim that their work reveals how the activity of SPOT is regulated by ceramide and propose a mechanism for how ergosterol might regulate SPT activity by favoring the monomeric over dimeric conformation. However, unequivocal assignment of density to a specific lipid is difficult in cryoEM maps of limited resolution. Thus, functional validation is needed to support these claims.

Similarly, the authors report the structure of SPOT in monomeric and dimer forms and argue that the monomeric form is the physiologically relevant one in yeast. However, such conclusions, while important, can and should be readily testable with mutants that can disrupt the interaction interface, and should thus favor the monomeric form of the enzyme.

I have some concerns on the structural assignment of the lipid molecules:

- The densities associated with the ergosterol lipids are weak and quite featureless. Further, there are numerous additional densities nearby of similar shape and intensity. It is not clear what criterion was used to assign these 3 specific densities to ergosterol and leave the others unassigned.
- Similarly, the density for the 44:0;4 ceramide is weak and discontinuous. Some connectivity is visible only at threshold levels where the noise density of the micelle becomes significant, and in all maps most of its long chain seems to be built outside of the density. (e.g. see image below)

While the accompanying biochemical analysis shows that ergosterol and ceramide lipids copurify with the SPOT protein, their structural assignment remains ambiguous.

I recommend the authors tone down their statements, trim some of the chains that are built and add disclaimers to the assignments. As mentioned above, that some functional studies would greatly strengthen the manuscript and would go a long way in supporting the assignment of these densities as these specific lipids. For example, the authors could mutate some of the coordinating residues and show that regulation by ceramide or sterol is ablated.

Additionally, some of the loops and peripheral helices are built into very weak density. The assignment of side chains and, in some cases, mainchain backbone is ambiguous. It would be more appropriate to build just the backbone trace of these regions, or maybe leave them as disconnected. See an example below.

Reviewer #2 (Remarks to the Author):

In this study, the authors focused on yeast SPOT and SPOTS complexes that are important for proper regulation of serine palmitoyltransferase (SPT), a rate limiting enzyme of sphingolipid biosynthesis. By using cryo-EM technique, the authors solved three-dimensional structure of SPOT and SPOTS complexes, and interaction of each component (Lcb1, Lcb2, Tsc3, Orm1, and Sac1) were studied in detail. The experiments are well done and the reviewers have no specific comments on the data currently presented; however, additional experiments are required to support the conclusion. This study describes how Orm1, Sac1, and Tsc3 are involved in the regulation of SPT based on the analysis of the three-dimensional structure of the complexes; however, point mutants and deletion mutants were not analyzed. Therefore, this study is lacking the experimental evidences regarding importance of each amino acid and domain structure in the interaction of each component and the in vivo regulation of SPT activity in SPOT and SPOTS complexes.

1. It is interesting that ceramide and ergosterol are present in the SPOT and SPOTS complexes. The fact that ceramide levels were decreased in Orm1-free SPT samples is also interesting from the viewpoint of ceramide-mediated regulation of SPT activity. The authors pointed out that ceramide association with SPOT complex are associated with the regulation of SPT activity via Orm, but no experimental evidence exists. The sensitivity to myriocin, sphingolipid profile in myriocin-treated cells, and the in vivo interaction between Orm protein and SPT should be examined by expressing point mutants for amino acids that are thought to be important for the interaction with ceramide.

2. Myriocin resistance is observed in SAC1-deleted strain, which suggests that Sac1 negatively regulates SPT activity (Nature (2010)463,1048). In this study, analysis of the cryo-EM structure of the SPOTS complex indicated the interaction between Sac1 and Lcb2. The interaction between Sac1 and Lcb2 is described in detail in this paper; however, the mechanism of how Sac1 affects SPT activity is not discussed. The authors should investigate how point mutations and deletion mutations that affect the interaction between Sac1 and Lcb2 affect the formation of the SPOTS complex and myriocin resistance in yeast. These experiments will provide strong evidence regarding the regulation of SPT activity by Sac1 through its interaction with Lcb2.

3. Phosphorylation of Orm proteins is a critical factor in whether SPT and Orm proteins form a complex. In the discussion section, the authors described the potential role of the phosphorylation on formation of the complexes. Is it possible to demonstrate by some simulation studies for observing the difference in complex formation between phosphorylated Orm1 and dephosphorylated one with SPT?

4. In this manuscript, Orm1 is used to analyze SPOT and SPOTS complexes. Based on sequence similarity to Orm1, it would be beneficial if the author discuss how Orm2 interacts in the SPOT and SPOTS complex and whether the phosphorylation of Orm2 also affects the complex formation in the same way as Orm1.

Reviewer #1

Ceramides are lipids that play a variety of vital roles in cellular homeostasis. Dysregulated synthesis of these sphingolipids can lead to catastrophic consequences and is associated with a variety of pathologies. Activity of the serine palmitoyltransferase (SPT) enzyme, which is responsible for initiating the rate-limiting step in sphingolipid biosynthesis, is modulated by its association with ORM/ORMDL proteins and by cellular levels of ceramides and sterols. SPT is highly conserved across species, and in humans it is a dimer that further associates with the regulatory partners ORMDL3 and ssSPTa. Activity of SPT enzymes is further regulated by multiple cellular factors, such as levels of sphingolipids or sterols, and phosphorylation. The structural bases for these diverse modes of regulation remain poorly understood. Here, the authors report the structures of the SPT-Orm-Tsc3 complex (SPOT) in dimer and monomer forms, as well as of that of SPOT in complex with PI4P phosphatase Sac1 (SPOTS complex). These structures show that in the yeast complex the dimer interface is reduced, which allows the SPOT complex to also exist in monomeric form. Further, they show that Sac1 only binds to the SPOT monomer and the structures show how its association is sterically incompatible with formation of a SPOT dimer. Finally, these structures reveal the presence of several non-protein densities associated with the complex, which the authors assign to 44:0;4 ceramide and ergosterol molecules respectively. Notably, the 44:0;4 ceramide density is coordinated between the Orm1 and Lcb1 subunits and blocks the SPT substrate channel, suggesting a novel regulatory mechanism of SPT activity by sphingolipids.

Overall, the work is interesting, and the structures reported here provide insights into the assembly and regulation of this important enzyme. However, the lack of functional validation of the structure-driven inferences limits the interest of the present manuscript. The authors take great care of placing their structures in context of past results and illustrate how the present work helps in rationalizing these data. However, all insights obtained from their work remain hypothetical and speculative. The present manuscript would be greatly strengthened by functional work where specific predictions are tested.

We thank the reviewer for her/his time and the overall positive evaluation of our manuscript. We agree that functional studies strengthen this study. A detailed description of the functional assays we have added is listed below.

For example, the authors claim that their work reveals how the activity of SPOT is regulated by ceramide and propose a mechanism for how ergosterol might regulate SPT activity by favoring the monomeric over dimeric conformation. However, unequivocal assignment of density to a specific lipid is difficult in cryoEM maps of limited resolution. Thus, functional validation is needed to support these claims.

To address the reviewer's concerns, we mutated Y485 or Y110 and tested the activity of the mutants in cells to directly evaluate the role of the two proposed gate-keeper residues in the ceramide mediated inhibition of the serine-palmitoyltransferase. We also introduced bulky phenylalanines in the Lcb2 amphipathic helix and the Orm1 TM1 to create a steric hinderance of ceramide binding. We also analyzed the sphingolipid levels in these mutants. All experiments are part of Figure 3 now.

We have also added data showing that deleting the N-terminal TM0a helix and the complete TM0 yields stable complexes and deletion of TM0a does indeed have an effect on the levels of 3-KS and LCBs in vivo.

In addition, we show that deleting the C-terminal β -hairpin of Sac1 indeed causes the loss of the interaction with Lcb2. We also show that this interaction is important for the activity of the complex in vivo.

All this functional data is now included in the revised manuscript.

Similarly, the authors report the structure of SPOT in monomeric and dimer forms and argue that the monomeric form is the physiologically relevant one in yeast. However, such conclusions,

while important, can and should be readily testable with mutants that can disrupt the interaction interface, and should thus favor the monomeric form of the enzyme.

As suggested by the referee we have now tested the presence of dimers in vivo using pulldown assays of differentially tagged Lcb1 subunits in diploid yeast cells. In our hands we are only able to co-purify minimal amounts of the differentially tagged Lcb1 subunit, suggesting that most of the SPOTS complex is present in its monomeric form in vivo.

I have some concerns on the structural assignment of the lipid molecules:

- The densities associated with the ergosterol lipids are weak and quite featureless. Further, there are numerous additional densities nearby of similar shape and intensity. It is not clear what criterion was used to assign these 3 specific densities to ergosterol and leave the others unassigned.

- Similarly, the density for the 44:0;4 ceramide is weak and discontinuous. Some connectivity is visible only at threshold levels where the noise density of the micelle becomes significant, and in all maps most of its long chain seems to be built outside of the density. (e.g. see image below)

While the accompanying biochemical analysis shows that ergosterol and ceramide lipids copurify with the SPOT protein, their structural assignment remains ambiguous.

I recommend the authors tone down their statements, trim some of the chains that are built and add disclaimers to the assignments. As mentioned above, that some functional studies would greatly strengthen the manuscript and would go a long way in supporting the assignment of these densities as these specific lipids. For example, the authors could mutate some of the coordinating residues and show that regulation by ceramide or sterol is ablated.

Additionally, some of the loops and peripheral helices are built into very weak density. The assignment of side chains and, in some cases, mainchain backbone is ambiguous. It would be more appropriate to build just the backbone trace of these regions, or maybe leave them as disconnected. See an example below.

We thank the reviewer for carefully analyzing our provided structural data. As rightfully mentioned, confident model building in the 3-4 Å range relies on complementary data, supporting the assignment of non-protein densities to specific ligands. Additionally, density-modification within Phenix helped us to increase the visibility of details; specifically for building ergosterol-molecules as shown in **Fig. 1A**. In contrast to X-ray crystallography, automated ligand assignment largely remains a developing field in the context of cryo-EM (EMERALD, <https://doi.org/10.1038/s41467-023-36732-5>). We used confidence maps from false-discovery-rate-thresholding (set to 0.01%) for potential ligand density selection and verified our assignment by mass-spectrometry in the case of the ergosterol and 44:0;4 ceramide (<https://doi.org/10.1107/S2052252518014434>), **Fig. 1B**. Other potential ligand-densities were left un-assigned due to weak signal-to-noise ratio. If requested, we provide the reviewers with our supporting maps.

Fig. 1: Local map quality improvement. **A** Phenix density modified map-model fit of ergosterol in our SPOTS structure, contoured at 2σ . **B** Focus on the map-model fit of 44:0;4 ceramide and SPOTS. Density modified map (2σ) in cyan and confidence map (0.01 % FDR threshold) in violet.

The reviewers noticed a decrease in map quality for the long-chain fatty acid of the modeled 44:0;4 ceramide. We rationalized this observation with the high degree of flexibility of the solvent-exposed long-chain fatty acid region, which results in mixed occupancy within the aligned particles in the final reconstructions. Additionally, the absence of interacting protein residues and detergent (GDN) of the mentioned region of the ceramide results in the observable densities. A recent preprint (<https://www.biorxiv.org/content/10.1101/2023.01.18.524529v2>) provides the novel tool “OccuPy” to estimate compositional heterogeneity from cryo-EM reconstructions. The heterogeneity leads to local attenuation of the reconstructed density, which the authors termed local scale. By using this tool, we show the extent of flexibility and / or partial occupancy within the overall reconstructions and specifically for the local ceramide densities.

These results support our assigned models and are now incorporated in the manuscript as supplementary figure 11.

Furthermore, we have re-evaluated and truncated the model residues within the transmembrane-regions of Tsc3 and Sac1 according to the reviewer's suggestions (Fig. 2).

Fig. 2: Local map quality improvement. **A** Dimer-Masked Tsc3-TM1 (3σ), Monomer Tsc3-TM1 (2σ), Monomer-Sac1 Tsc3-TM1 (2σ), Monomer-Sac1, Sac1-TM1-2 (2.6σ).

Reviewer #2

In this study, the authors focused on yeast SPOT and SPOTS complexes that are important for proper regulation of serine palmitoyltransferase (SPT), a rate limiting enzyme of sphingolipid biosynthesis. By using cryo-EM technique, the authors solved three-dimensional structure of SPOT and SPOTS complexes, and interaction of each component (Lcb1, Lcb2, Tsc3, Orm1, and Sac1) were studied in detail. The experiments are well done and the reviewers have no specific comments on the data currently presented; however, additional experiments are required to support the conclusion. This study describes how Orm1, Sac1, and Tsc3 are involved in the regulation of SPT based on the analysis of the three-dimensional structure of the complexes; however, point mutants and deletion mutants were not analyzed. Therefore, this study is lacking the experimental evidences regarding importance of each amino acid and domain structure in the interaction of each component and the in vivo regulation of SPT activity in SPOT and SPOTS complexes.

We would like to thank the referee for his/her overall positive evaluation of our manuscript. We have now included the analysis of several point mutations clarifying many of the raised

questions. A detailed description of our additional experiments is below.

1. It is interesting that ceramide and ergosterol are present in the SPOT and SPOTS complexes. The fact that ceramide levels were decreased in Orm1-free SPT samples is also interesting from the viewpoint of ceramide-mediated regulation of SPT activity. The authors pointed out that ceramide association with SPOT complex are associated with the regulation of SPT activity via Orm, but no experimental evidence exists. The sensitivity to myriocin, sphingolipid profile in myriocin-treated cells, and the in vivo interaction between Orm protein and SPT should be examined by expressing point mutants for amino acids that are thought to be important for the interaction with ceramide.

We have now added functional data to the manuscript showing the importance of the two LCB2 gatekeeper tyrosine residues in SPT activity (See response to Reviewer #1). These residues are important to coordinate the ceramide in the structure and we now investigate their importance for SPT activity and stability.

It was previously shown that the deletion of TM1 of Lcb1 leads to the dissociation of the Orm proteins from the SPT complex. However, this has only a small effect on the activity of the SPT (DOI: 10.1016/j.bbajp.2018.11.007). To analyze the role of ergosterol binding to the SPOTS complex, we analyzed deletions of the TM0a and the entire TM0 helix which still allows purification of the complex. Yeast cells harboring these mutations have slightly reduced levels of 3-KS and LCBs suggesting that this could indeed have a regulatory role. We also generated bulky hydrophobic mutations in Lcb2 and Orm1 to displace the ceramide from the complex. A combination of these mutations led to a higher resistance of these strains against the SPT inhibitor myriocin, suggesting overall higher SPT activity.

2. Myriocin resistance is observed in SAC1-deleted strain, which suggests that Sac1 negatively regulates SPT activity (Nature (2010)463,1048). In this study, analysis of the cryo-EM structure of the SPOTS complex indicated the interaction between Sac1 and Lcb2. The interaction between Sac1 and Lcb2 is described in detail in this paper; however, the mechanism of how Sac1 affects SPT activity is not discussed. The authors should investigate how point mutations and deletion mutations that affect the interaction between Sac1 and Lcb2 affect the formation of the SPOTS complex and myriocin resistance in yeast. These experiments will provide strong evidence regarding the regulation of SPT activity by Sac1 through its interaction with Lcb2.

We addressed this point by generating a mutation lacking the C-terminal β -hairpin of Sac1. This mutant does not bind to the SPOT complex anymore as highlighted by co-immunoprecipitation experiments that are now part of figure 4. While we agree that the myriocin resistance of a SAC1 deletion is interesting we believe that this is a rather indirect readout for the role of Sac1 in the SPOTS complex. From the literature it is clear that the deletion of SAC1 will influence the lipid exchange cycle between ergosterol and phosphatidylinositol-4-phosphate. This can have multiple effects on the membrane structure, the uptake rate of myriocin etc. We therefore think that directly measuring long chain base levels and 3-KS levels in the different SAC1 mutants is more informative. As shown previously, deletion of SAC1 results in increased levels of long chain bases (doi: 10.1074/jbc.M808325200.). We now show that expressing a C-terminal deletion of Sac1 can partially rescue this phenotype. A detailed discussion of these phenotypes is now part of the manuscript.

3. Phosphorylation of Orm proteins is a critical factor in whether SPT and Orm proteins form a complex. In the discussion section, the authors described the potential role of the phosphorylation on formation of the complexes. Is it possible to demonstrate by some simulation studies for observing the difference in complex formation between phosphorylated Orm1 and dephosphorylated one with SPT?

While we agree with the referee, we were not able to simulate the effects of phosphorylation

on the interaction of Orm1 and the rest of the SPOTS complex. What is evident in our structures is that the phosphorylation sites are in a very accessible region of the complex. In a recent preprint from the lab we show that the effect of an Orm2 deletion has stronger effects on the lipid levels compared to an Orm1 deletion (<https://doi.org/10.1101/2023.03.29.534722>). This could either be dependent on the overall protein amounts in the cell or on a different regulatory mechanism. In this preprint we propose that the intracellular localization of the complexes is crucial for the regulation. This would thus be independent from structural features of the complex. However, we used an Orm1-3A mutant to stabilize the SPOTS complex. It will be interesting in the future to try to purify complexes harboring Orm1-3D mutants but so far, this has been described to destabilize the complex.

4. In this manuscript, Orm1 is used to analyze SPOT and SPOTS complexes. Based on sequence similarity to Orm1, it would be beneficial if the author discuss how Orm2 interacts in the SPOT and SPOTS complex and whether the phosphorylation of Orm2 also affects the complex formation in the same way as Orm1.

As suggested, we have now included a discussion on how Orm2 interacts with the SPOTS complex. In fact, we have preliminary structural data of the Orm2 containing SPOT complex and the overall architecture is highly similar (Fig. 3).

Fig 3: Structure of the SPOT-Orm2 dimer. **A** Cryo-EM density of the SPOT-Orm2 dimer at 3.3 Å. **B** Superposition and $C\alpha$ RMSD values for the overall complex and the Orm subunits within the SPOT-dimer complexes involving Orm1 (red) and Orm2 (blue).

REVIEWERS' COMMENTS

Reviewer #1 (Remarks to the Author):

The authors have addressed my concerns and I appreciate the additional work that greatly strengthens their conclusions.

I would recommend that all the cryoEM maps (incl. the density modified maps) are made accessible to the public, so that interested scientists can directly evaluate the data utilized to build models and assign densities.

Reviewer #2 (Remarks to the Author):

The authors adequately revised the manuscript to address the reviewer's comments.